# Evaluation of the Impact of the Urgent Cancer Care Clinic on Emergency Department Visits, Primary Care Clinician Visits, and Hospitalizations in Winnipeg, Manitoba

Katie Galloway [1,2], Pascal Lambert [1,3], Eric J. Bow [4,5,6], Piotr Czaykowski [2,4,5], Tunji Fatoye [7,8], Benjamin Goldenberg [4,5], Mark Kristjanson [7], Harminder Singh [2,3,5], Oliver Bucher [1] and Kathleen Decker [1,2,3,*]

1 Department of Epidemiology and Cancer Registry, CancerCare Manitoba, 675 McDermot Avenue, Winnipeg, MB R3E 0V9, Canada; kgalloway@cancercare.mb.ca (K.G.); plambert@cancercare.mb.ca (P.L.); obucher@cancercare.mb.ca (O.B.)
2 Department of Community Health Sciences, Rady Faculty of Health Sciences, Max Rady College of Medicine, University of Manitoba, 750 Bannatyne Avenue, Winnipeg, MB R3E 0W2, Canada; pczaykowski@cancercare.mb.ca (P.C.); harminder.singh@umanitoba.ca (H.S.)
3 Paul Albrechtsen Research Institute CancerCare Manitoba, 675 McDermot Avenue, Winnipeg, MB R3E 0V9, Canada
4 Department of Medical Oncology and Hematology, CancerCare Manitoba, 675 McDermot Avenue, Winnipeg, MB R3E 0V9, Canada; ebow@cancercare.mb.ca (E.J.B.); bgoldenberg@cancercare.mb.ca (B.G.)
5 Department of Internal Medicine, Rady Faculty of Health Sciences, Max Rady College of Medicine, University of Manitoba, 820 Sherbrook Street, Winnipeg, MB R3A 1R9, Canada
6 Department of Medical Microbiology and Infectious Diseases, Rady Faculty of Health Sciences, Max Rady College of Medicine, University of Manitoba, 745 Bannatyne Avenue, Winnipeg, MB R3E 0J9, Canada
7 Department of Primary Care Oncology, Cancer Care Manitoba, 675 McDermot Avenue, Winnipeg, MB R3E 0V9, Canada; tfatoye@sogh.mb.ca (T.F.); mkristjanson@cancercare.mb.ca (M.K.)
8 Department of Family Medicine, Rady Faculty of Health Sciences, Max Rady College of Medicine, University of Manitoba, 750 Bannatyne Avenue, Winnipeg, MB R3E 0W2, Canada
* Correspondence: kdecker@cancercare.mb.ca; Tel.: +1-204-390-3912

**Abstract:** The urgent cancer care (UCC) clinic at CancerCare Manitoba (CCMB) opened in 2013 to provide care to individuals diagnosed with cancer and serious blood disorders experiencing complications from the underlying disorder or its treatment. This study examined the impact of the UCC clinic on other health care utilization in Winnipeg, Manitoba, Canada. An interrupted time series study design was used to compare the rates of emergency department (ED) visits, primary care clinician (PCC) visits, and hospitalizations from 1 January 2010 to 31 December 2015. Rates of ED visits were also stratified by ED location, severity, and cancer type. We found a 6% (95% CI 1.00–1.13, *p*-value = 0.0389) increase in PCC visits, a 7% (95% CI 0.99–1.15, *p*-value = 0.0737) increase in hospitalizations, a 4% (95% CI 0.86–1.08, *p*-value = 0.5053) decrease in the rate of ED visits, and a 3% (95% CI 0.92–1.17, *p*-value = 0.5778) increase in the rate of ED visits during the UCC clinic hours after the UCC clinic opened. The implementation of the UCC clinic had minimal impact on health care utilization. Future work should examine the impact of the UCC clinic on other aspects of healthcare utilization (e.g., number of tests ordered and time spent waiting in CCMB's main clinics) and patient quality of life and patient and health care provider experience.

**Keywords:** cancer; urgent care clinic; health care utilization; emergency department visits; primary care clinician visits; hospitalizations; epidemiological studies

## 1. Introduction

After a cancer diagnosis, individuals experience higher emergency department (ED) use compared with individuals who do not have cancer as a result of needing to seek care for symptoms related to their cancer diagnosis and its treatment [1–3]. These cancer-related ED visits often result in increased hospital admission rates compared with non-cancer-related ED visits [4–7]. The most common reasons for attending an ED among

individuals diagnosed with cancer are pain, fatigue, respiratory complaints, gastrointestinal complaints, fever, and other infection-related issues [1,4,7–13]. The most common reasons for hospitalization among individuals diagnosed with cancer include anemia, neutropenia, and sepsis [14]. EDs often lack cancer-specific resources needed to support this complex population of individuals. Therefore, EDs may not always be the most appropriate setting to provide care for individuals diagnosed with cancer [5,13]. In recent years, urgent cancer care (UCC) clinics within cancer centers have become popular alternatives to traditional EDs for individuals diagnosed with cancer who are experiencing acute complications related to the underlying cancer diagnosis or its treatment [15–26]. Planning for the UCC clinic at CancerCare Manitoba (CCMB) began in 1998 in response to a gap in care identified by clinicians. It was noted that there was a need for a service to supplement the care provided by the regular clinics at CCMB. If regular clinics at CCMB were fully booked, a patient arriving for a clinic appointment with a new incidental complex problem could not be given the time needed for appropriate problem solving without disrupting clinic workflow and the time needed for other patients. This also applied to an individual's primary care clinician's (PCC) practice. Individuals with new problems requiring a full assessment therefore had to be rescheduled for another time, resulting in a delay of care that posed a safety risk for the individual. Furthermore, new problems often developed at times when regular clinics or PCC practices were not open. Therefore, timely access to care by professionals with experience in cancer-related problems was one of the primary reasons that led to the implementation of the UCC clinic at CCMB in November 2013. It has since been shown to provide timely care, with reduced wait times compared with an ED [27]. Additionally, the UCC clinic also provides increased convenience and familiarity for individuals diagnosed with cancer and their families that may result in fewer delays in seeking care, resulting in safer care. Since the UCC clinic at CCMB is located within CCMB's MacCharles Unit, where its clinicians have access to the patient's oncology team and CCMB's electronic medical record and follow-up care after an UCC clinic visit may be provided by an individual's PCC, it was hypothesized that the UCC clinic offers the unique opportunity to provide more coordinated care between an individual's oncology team and their PCC.

Reduced use of other health care services (ED visits, hospitalizations, and PCC visits) was not a direct goal of the implementation of the UCC clinic. However, it was thought that the UCC clinic may have an impact on other health care utilization if individuals chose to seek care from the UCC clinic instead of from a traditional ED or PCC. Additionally, it was thought that the contextual care provided by the UCC clinic may result in better symptom management and fewer ED visits or hospitalizations. Information about the impact of UCC clinics on other health care utilization is limited. Prior research has shown that ED visits and hospitalizations may decrease after the implementation of a UCC clinic, but the results are not consistent across jurisdictions [15,20–22]. The implementation of an ED designed for individuals diagnosed with cancer in Seoul, Korea resulted in a significant reduction in admissions to inpatient units [15]. However, a UCC clinic at a cancer center in Texas did not lead to any significant decreases in hospitalizations [20]. The same study in Texas found a significant reduction in ED visits after the UCC clinic opened [20]. Similarly, an oncology extended care clinic (OECC) in Connecticut resulted in a non-significant decrease in hospitalizations and a significant decrease in ED visits [22]. In contrast, after the opening of a cancer symptom management clinic in a cancer center in Pennsylvania, ED visits increased [21]. With limited research in this area, it is still unclear how a UCC clinic impacts other health care utilization. Since each UCC clinic is different in the way the clinic is designed, where the clinic is located, and the individuals seen by the clinic, it is important to understand the impact of the UCC clinic on health care utilization in each unique setting so that other health care systems can learn from UCC clinics that are implemented in similar settings. The objective of this study was to evaluate the impact of the UCC clinic at CCMB on ED visits, PCC visits, and hospitalizations in Winnipeg, Manitoba, Canada from November 2013 to December 2015.

## 2. Materials and Methods

### 2.1. Setting

As the provincial cancer agency in Manitoba, CCMB is responsible for providing clinical services to all Manitobans diagnosed with cancer and blood disorders. The UCC clinic operates within CCMB's main site located in downtown Winnipeg, directly adjacent to the Health Sciences Centre (HSC), Manitoba's largest tertiary health care facility. There were six EDs operating throughout Winnipeg during the study period, including an ED at the HSC. There were also EDs at St. Boniface Hospital (Manitoba's second largest tertiary health care facility), Seven Oaks General Hospital, Grace Hospital, Concordia Hospital, and Victoria General Hospital.

### 2.2. Intervention

After many years of planning and concept development, the UCC clinic at CCMB opened on 4 November 2013. The UCC clinic was implemented to supplement the care provided in regular clinics at CCMB. The goal of the UCC clinic is to provide timely, comprehensive, coordinated, and contextual care for individuals diagnosed with cancer and serious blood disorders who are experiencing complications of the underlying disorder or its treatment. The UCC clinic provides convenience for individuals diagnosed with cancer by providing direct access to timely care without having to wait long hours in an ED. The UCC clinic care team has access to the patient's oncology team and CCMB's electronic medical record, so they are able to coordinate care with the oncology team. Since the UCC clinic is staffed by primary care health care providers (family physicians and nurse practitioners (NPs)), many of whom have received focused specialized training in oncology and blood disorders, they are able to provide comprehensive and contextual care, because the UCC clinic team understands cancer, its complications, and how the individual came to the point of needing the kind of care the UCC clinic can provide. They provide the expert care needed for managing complications of the underlying diagnosis or its treatment in individuals diagnosed with cancer in a prompt manner. While the staff in the UCC clinic are available to manage sudden and unexpected complications, they do not provide guidance on treatment selection. Regular care and treatment decisions are provided by hematologists and oncologists through scheduled visits at regular clinics throughout CCMB. The UCC clinic contains exam bays and treatment stretchers to provide the care needed by the individual.

Individuals must be receiving treatment or follow-up care from a CCMB clinic in order to be eligible to attend the UCC clinic and can be referred to the UCC clinic by their oncology team, PCC, or by self-referral. Upon arrival at the UCC clinic, individuals are registered, assessed and triaged, and seen by a family physician (or NP) in oncology (FPO). The individual may undergo investigations such as diagnostic imaging, microbial cultures, or blood tests. After a medical assessment by the FPO/NP, immediate treatment such as intravenous fluids, blood products, antimicrobial agents, or other medications may be provided as appropriate. The individual may be discharged home with additional follow-up care provided by their oncology team or their PCC, they may be referred to an outpatient subspecialty service for further urgent care management (i.e., interventional respirology for urgent outpatient thoracentesis or endoscopic retrograde cholangiopancreatography for biliary decompression), or they may be sent to an acute care hospital service such as an ED or directly to an inpatient hospital-based service.

### 2.3. Study Design

An interrupted time series (ITS) study design [28] was used to compare the rate of ED visits, PCC visits, and hospitalizations prior to and after the opening of the UCC clinic. The pre-intervention time period, prior to the opening of the UCC clinic, was from 1 January 2010 to 3 November 2013. The post-intervention time period, after the UCC clinic was implemented, was from 4 November 2013 to 31 December 2015. With this study design, the rates after the UCC clinic opened were compared with the rates that were expected

to occur if the UCC clinic had not been implemented. This study was approved by the University of Manitoba's Health Research Ethics Board (HS20816; H2017:167), Manitoba Health's Health Information and Privacy Committee (2017/2018–08), CCMB's Research and Resource Impact Committee (2017-13), and Winnipeg Regional Health Authority's Research Access and Approval Committee (now known as Shared Health's Approval Committee for Privacy, Impact and Access in Research) (2017-044).

### 2.4. Data Sources and Study Population

The Manitoba Cancer Registry (MCR) was used to identify individuals diagnosed with invasive cancer (excluding non-melanoma skin cancer) between 2009 and 2015 who were living in Winnipeg, were 18 years of age or older at the time of diagnosis, and were receiving treatment or follow-up care at CCMB. Follow-up care included anyone being seen at CCMB, so this could include individuals under a watchful waiting period or those who were waiting to start treatment at CCMB, as long as they had seen a clinician at CCMB. The analysis of health care utilization was restricted to health care visits that occurred within six months of diagnosis; only visits that occurred between 2010 to 2015 were included. Therefore, only individuals diagnosed between 1 July 2009 and 31 December 2015 contributed person-days to the analysis. The MCR is a population-based registry, legally mandated to collect comprehensive information about all cancer cases in Manitoba. The MCR has been shown to contain high quality data and has consistently achieved the gold standard of certification according to the North American Association of Central Cancer Registries [29]. The MCR uses disease site groupings according to the *International Classification of Diseases for Oncology, Third Edition* (ICD-0-3). The Medical Claims database was used to determine PCC visits. This database is maintained by Manitoba Health and is populated from claims submitted for reimbursement of services provided by health care providers. The Hospital Discharge Abstracts Database collects information on all hospital admissions for Manitoba residents and was used to determine hospitalizations in this study. Previously, these data sources have been validated for accuracy and have been used to study many health outcomes [30,31]. The Winnipeg Regional Health Authority's Admissions, Discharge and Transfer and E-triage data, and Emergency Department Information System databases were used to identify ED visits. The 5-level Canadian triage and acuity scale (CTAS) scores (level 1: resuscitation; level 2: emergent; level 3: urgent; level 4: less urgent; level 5: non-urgent) was used to assess the severity of an individual's condition at each ED visit.

### 2.5. Outcomes

All ED visits, only ED visits during the UCC clinic hours of operation, PCC visits, and hospitalizations in the six months after an individual's cancer diagnosis were included as the outcomes in this study. This six-month time period was chosen because our previous study showed that the rate of ED visits was highest immediately after diagnosis and the rate of UCC visits increased during the first 4 months after diagnosis and then decreased over time [27]. The aggregate monthly counts were converted to monthly rates (per person-days) by dividing the number of visits in a month by the total number of days that individuals were eligible for the study within that month. These monthly rates were used in the analyses. Non-cancer related ED visits were excluded from the analyses and are described in a previous paper [27]. Rates of ED visits during the UCC clinic hours of operation were also stratified by ED location (HSC, St. Boniface Hospital, Seven Oaks General Hospital, Grace Hospital, Concordia Hospital, and Victoria General Hospital), CTAS score (1 to 2 and 3 to 5), and cancer type (breast, digestive, lung, genitourinary, and hematologic cancers). These cancer types were grouped to ensure large enough sample sizes for the analysis and were chosen based on discussions with clinicians (Table S1). These subgroup analyses were chosen in order to provide clinicians with the necessary information in order to explain the impact of the UCC clinic on ED visits. A sensitivity analysis was conducted that included all ED visits, only ED visits during the UCC clinic hours of operation, PCC visits, and

hospitalizations during the first year after an individual's cancer diagnosis. This sensitivity analysis was chosen to ensure the study did not miss any impact on health care use beyond the initial six months after diagnosis, while individuals were still receiving treatment.

### 2.6. Statistical Analyses

For each outcome, a generalized linear model (GLM) (Poisson, quasi-Poisson, negative binomial, gamma, or inverse gaussian) or a weighted linear model were considered for the analysis. For the GLMs, the observed mean and variance in the data were plotted, along with the predicted mean and variance of each proposed model in order to determine model fit [32]. To assess the uniformity of residuals and dispersion of the data, scaled quantile residual plots were examined [33]. For the linear model, scaled residual plots were used to determine model fit. The terms included in each model were a variable indicating the month of the year in order to account for seasonality, a variable indicating the time (in months) since the start of the study period, and a binary variable indicating the intervention period (0 prior to the implementation of the UCC clinic and 1 after the UCC clinic opened). A GLM using the negative binomial distribution provided the best model fit for all outcomes and was therefore used for all analyses. Splines were used to describe non-linear time and seasonality trends observed in the data [34]. Using the chosen GLM, the McFadden pseudo R-squared [35] was compared between models in the model building process. The model with the higher McFadden pseudo R-squared was selected.

Using the chosen regression model for each outcome, predicted values were calculated to describe the smoothed trend in the observed data, referred to as the fitted values. For the period after the UCC clinic was implemented, counterfactual predictions were calculated under the assumption that the UCC clinic did not open. The observed, fitted, and counterfactual values were then plotted. When examining the plot, if the counterfactual values did not match what was seen in the baseline trend (the observed data in the period before the UCC clinic was implemented), then the models were simplified by reducing the number of degrees of freedom used for the splines until the counterfactual values agreed with the baseline trend. Since the UCC clinic opened on 4 November 2013, this month was removed from the analyses. Ratios between the fitted and counterfactual values, along with 95% confidence intervals (CI) of the ratio, were calculated for each outcome. Data manipulation and organization was completed using SAS (SAS Institute Inc., Cary, NC, USA). R software (R Foundation for Statistical Computing, Vienna, Austria) was utilized to conduct the analyses using the haven, splines, Hmisc, lattice, MASS, ggplot2, car, DHARMa, and multcomp packages.

## 3. Results

### 3.1. Characteristics of the Study Cohort

The study cohort contained 18,800 individuals. Table 1 contains characteristics of the cohort stratified into populations before and after the UCC clinic opened. There were no differences between individuals diagnosed before and after the opening of the UCC clinic in terms of age at diagnosis (*p*-value = 0.0929), sex (*p*-value = 0.1234), stage at diagnosis (*p*-value = 0.5551), and radiation therapy received within six months of diagnosis (*p*-value = 0.0883). There were differences between individuals diagnosed before and after the opening of the UCC clinic for cancer type (*p*-value = 0.0248), systemic therapy received within six months of diagnosis (*p*-value $\leq$ 0.0001), and surgery received within six months of diagnosis (*p*-value $\leq$ 0.0001).

**Table 1.** Characteristics of individuals in the study cohort by UCC status, 1 July 2009 to 31 December 2015.

| Characteristic | Before UCC Clinic (N = 12,323) | After UCC Clinic (N = 6477) | *p*-Value |
|---|---|---|---|
| Median age at diagnosis (IQR) | 66 (57–76) | 67 (58–76) | 0.0929 |
| Sex (%) | | | |
|   Female | 54.3 | 53.1 | |
|   Male | 45.7 | 46.9 | 0.1234 |
| Cancer type (%) | | | |
|   Breast | 18.5 | 16.9 | |
|   Digestive | 19.4 | 19.8 | |
|   Lung | 12.3 | 13.7 | |
|   Genitourinary | 23.7 | 23.6 | |
|   Hematologic | 10.3 | 10.4 | |
|   Other | 15.8 | 15.6 | 0.0248 |
| Stage at diagnosis (%) | | | |
|   I | 26.2 | 25.5 | |
|   II | 23.2 | 23.3 | |
|   III | 17.4 | 17.1 | |
|   IV | 21.0 | 21.9 | |
|   Unknown or Not Applicable | 12.3 | 12.1 | 0.5551 |
| Systemic therapy within six months of diagnosis (%) | | | |
|   Yes | 42.7 | 46.3 | |
|   No | 57.3 | 53.7 | <0.0001 |
| Radiation therapy within six months of diagnosis (%) | | | |
|   Yes | 24.7 | 25.9 | |
|   No | 75.3 | 74.1 | 0.0883 |
| Surgery within six months of diagnosis (%) | | | |
|   Yes | 53.9 | 47.9 | |
|   No | 46.1 | 52.1 | <0.0001 |

Abbreviations: IQR—interquartile range; UCC—urgent cancer care.

Between 2010 and 2015, this cohort experienced 85,357 PCC visits during the first six months after diagnosis, 51,298 PCC visits before the UCC clinic opened, and 34,059 PCC visits after the UCC clinic opened. There were 15,278 hospitalizations during the first six months after diagnosis, 9871 before, and 5407 after the UCC clinic opened. There were 11,325 ED visits during the first six months after diagnosis, 7372 visits prior to the UCC clinic opening, and 3953 visits after the opening of the UCC clinic. There were 4614 ED visits during the UCC clinic hours of operation in the six months after diagnosis. Additionally, there were 1319 UCC clinic visits during the first six months after diagnosis. There were 1052 ED visits during UCC hours of operation of CTAS score 1 to 2 and 3483 ED visits during UCC hours of operation of CTAS score 3 to 5. For individuals diagnosed with breast, digestive, lung, genitourinary, and hematologic cancers, there were 456, 1291, 867, 850, and 540 ED visits during the UCC clinic hours of operation in the six months after diagnosis, respectively.

For this cohort, there were 1370, 1100, 565, 534, 448, and 597 ED visits during the UCC clinic hours of operation in the six months after diagnosis at the HSC, St. Boniface Hospital, Seven Oaks General Hospital, Grace Hospital, Concordia Hospital, and Victoria General Hospital, respectively. During the post-intervention period, after the UCC opened, there were 554 visits at the HSC's ED during the UCC clinic hours. Of those visits, 50 individuals also had a UCC clinic visit on the same day. There were 354 visits at St. Boniface Hospital's ED during the post-intervention time period and less than five individuals had a UCC

clinic visit on the same day. There were no individuals who had a UCC clinic visit on the same day as having an ED visit at any of the remaining four EDs in Winnipeg.

### 3.2. PCC Visits

The ratios and 95% CI for PCC visits, hospitalizations, and ED visits are provided in supplemental data (Table S2). Figure 1a displays the observed PCC rates over time (N = 85,357 PCC visits). Shown in this figure are the fitted PCC rates based on the regression model and the counterfactual PCC rates that were expected if there had been no changes to the health care system (i.e., if the UCC clinic had not been implemented) during the post-intervention period. The 95% CI of the ratio of the fitted value to the counterfactual value is also provided in this figure (red shaded area). There was a 6% increase in PCC visits after the UCC clinic opened (RR = 1.06, 95% CI 1.00–1.13, *p*-value = 0.0389).

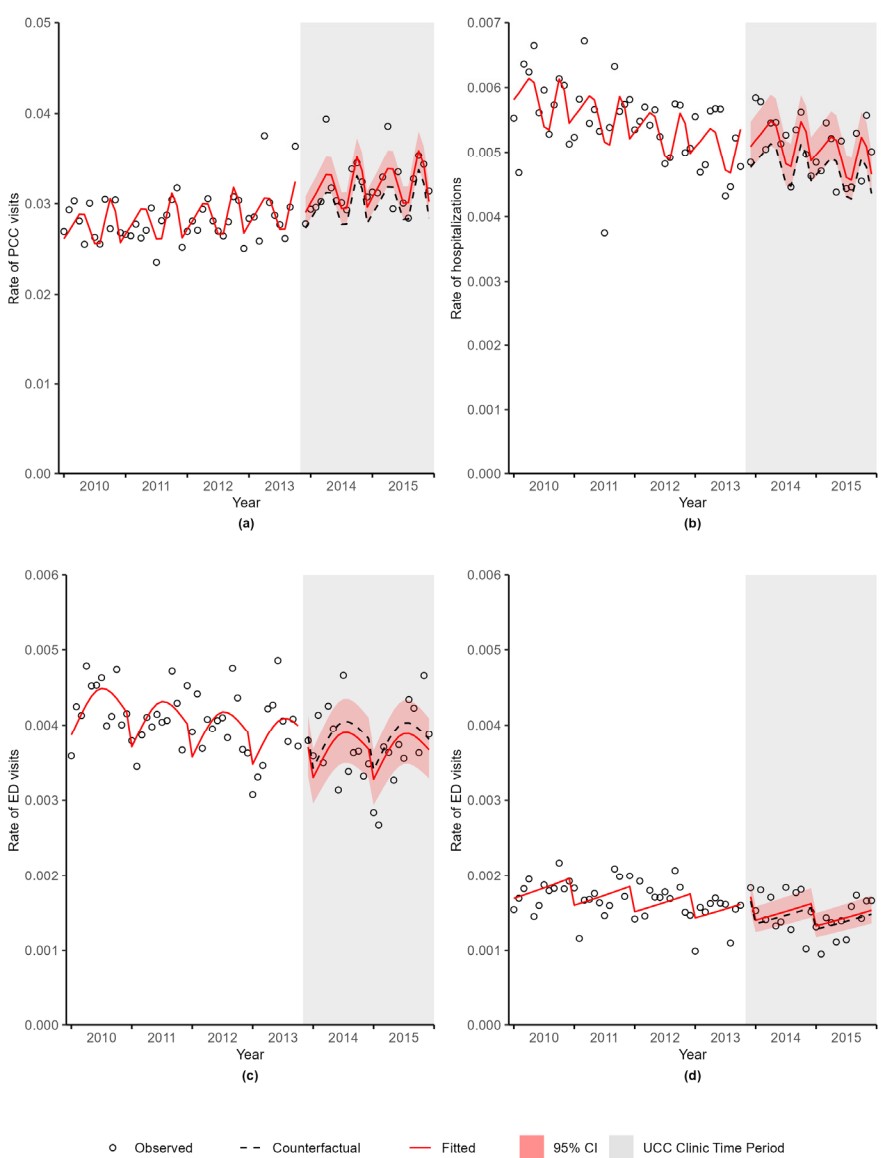

**Figure 1.** Rate (per person-days) of (**a**) primary care clinician visits, (**b**) hospitalizations, (**c**) emergency department visits, and (**d**) emergency department visits during the urgent cancer care clinic hours of operation by month, Winnipeg, Manitoba. Abbreviations: ED—emergency department; UCC—urgent cancer care; PCC—primary care clinician; CI—confidence interval.

### 3.3. Hospitalizations

Figure 1b shows a 7% increase in hospitalizations (N = 15,278 hospitalizations) after the UCC clinic was implemented (RR = 1.07, 95% CI 0.99–1.15, *p*-value = 0.0737).

### 3.4. ED Visits

Figure 1c displays the ED rates over time (N = 11,325 ED visits). There was a 4% decrease in the rate of ED visits after the opening of the UCC clinic (RR = 0.96, 95% CI 0.86–1.08, *p*-value = 0.5053). Figure 1d shows the rate of ED visits during the UCC clinic hours of operation (N = 4614 ED visits during UCC clinic hours of operation). There was an increase of 3% in the rate of ED visits during the UCC clinic hours after the UCC clinic opened (RR = 1.03, 95% CI 0.92–1.17, *p*-value = 0.5778).

### 3.5. ED Visit Subgroup Analyses

Figure 2 shows the observed, fitted, and counterfactual rates of ED visits during the UCC clinic hours of operation stratified by ED location. There was a 32% increase in the rate of ED visits at the HSC (the ED adjacent to the UCC clinic) (N = 1370 ED visits at HSC) (Figure 2a) after the implementation of the UCC clinic (RR = 1.32, 95% CI 1.00–1.74, *p*-value = 0.0500). There was a 31% decrease in the rate of ED visits at St. Boniface Hospital (N = 1100 ED visits at St. Boniface Hospital) after the UCC clinic opened, which can be seen in Figure 2b (RR = 0.69, 95% CI 0.54–0.89, *p*-value = 0.0041). There was a 28% decrease in the rate of ED visits at Seven Oaks General Hospital (N = 565 ED visits at Seven Oaks General Hospital) (RR = 0.72, 95% CI 0.49–1.04, *p*-value = 0.0818), a 5% increase in the rate of ED visits at Grace Hospital (N = 534 ED visits at Grace Hospital) (RR = 1.05, 95% CI 0.75–1.47, *p*-value = 0.7719), and a 29% increase in the rate of ED visits at Concordia Hospital (N = 448 ED visits at Concordia Hospital) (RR = 1.29, 95% CI 0.88–1.90, *p*-value = 0.1913) (Figure 2c–e, respectively). Due to unstable rates and poor model fit, an analysis was not possible for the Victoria General Hospital.

The observed, fitted, and counterfactual rates of ED visits during the UCC clinic hours of operation by CTAS score is provided in Figure S1 in supplemental data. There was a 20% decrease in the rate of ED visits during the UCC hours of operations for CTAS scores 1 to 2 (N = 1052 ED visits of CTAS score 1 to 2) (RR = 0.80, 95% CI 0.63–1.01, *p*-value = 0.0641) and a 10% increase for CTAS scores 3 to 5 (N = 3483 ED visits of CTAS score 3 to 5) (RR = 1.10, 95% CI 0.95–1.27, *p*-value = 0.2007). The figures for observed, fitted, and counterfactual rates of ED visits during the UCC clinic hours of operation for breast cancer, digestive cancers, lung cancer, and genitourinary cancers are provided in supplemental data (Figure S2). There was a 10% increase in the rate of ED visits for breast cancer (N = 456 ED visits for breast cancer) (RR = 1.10, 95% CI 0.76–1.60, *p*-value = 0.6103), a 1% decrease in the rate of ED visits for digestive cancers (N = 1291 ED visits for digestive cancers) (RR = 0.99, 95% CI 0.79–1.23, *p*-value = 0.9136), a 13% increase in the rate of ED visits for lung cancer (N = 867 ED visits for lung cancer) (RR = 1.13, 95% CI 0.84–1.51, *p*-value = 0.4272), and a 15% decrease in the rate of ED visits for genitourinary cancers (N = 850 ED visits for genitourinary cancers) (RR = 0.85, 95% CI 0.64–1.14, *p*-value = 0.2890). Due to unstable rates and poor model fit, analysis was not possible for hematologic cancers.

### 3.6. Sensitivity Analysis

The sensitivity analysis looking at the rates of health care utilization within one year after diagnosis found no substantive difference from the primary results.

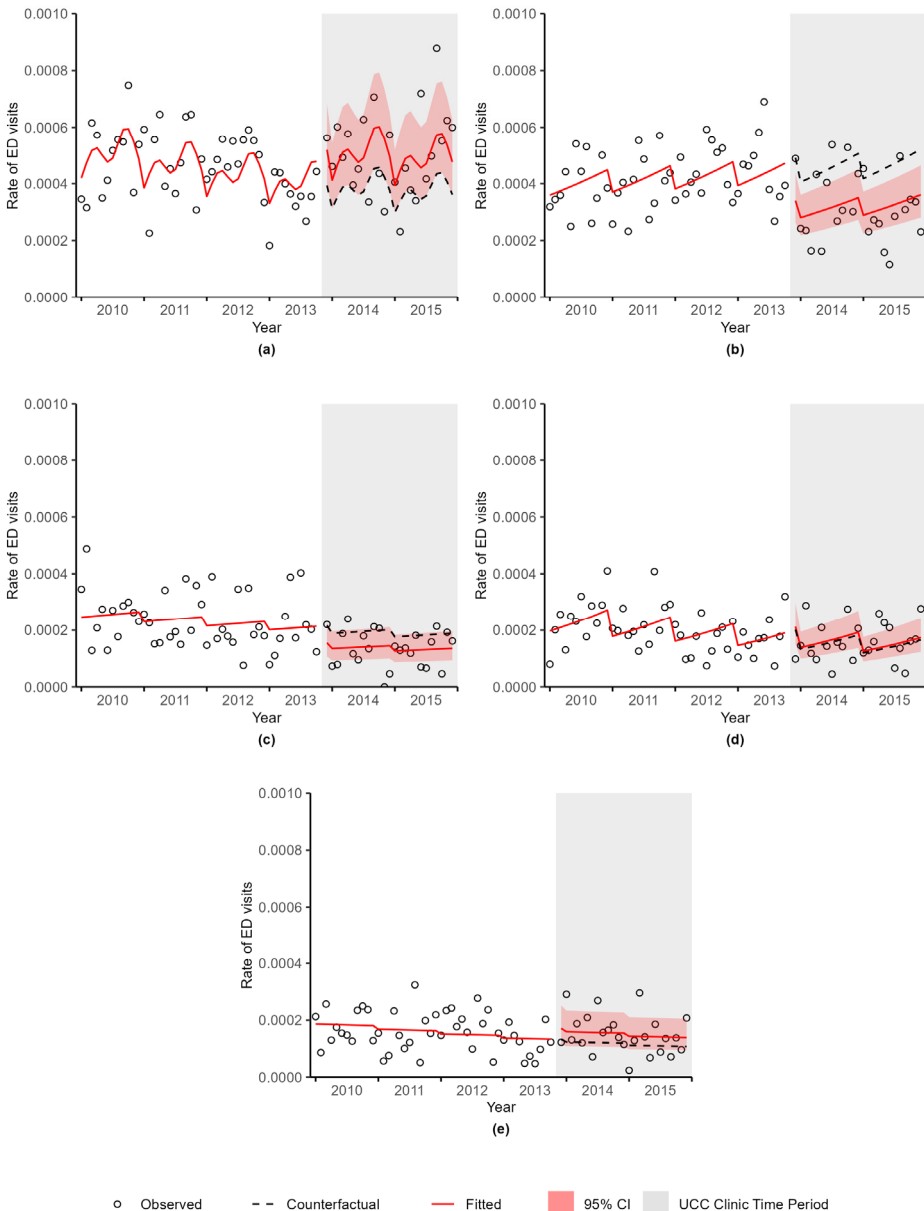

Figure 2. Rate (per person-days) of emergency department visits during the urgent cancer care clinic hours of operation at (**a**) Health Sciences Centre, (**b**) St. Boniface Hospital, (**c**) Seven Oaks General Hospital, (**d**) Grace Hospital, and (**e**) Concordia Hospital by month, Winnipeg, Manitoba.

## 4. Discussion

### 4.1. Main Findings

There was a 6% increase in the rate of PCC visits after the opening of the UCC clinic. This result was not surprising, since individuals are often discharged from the UCC clinic with follow-up care provided by their PCC. The UCC clinic offers the opportunity to provide more coordinated care between an individual's oncology team and their PCC. It is possible that an individual may not be satisfied with the care received and might see more than one doctor for the same illness [36]. However, our previous research showed that over 70 percent of individuals in the cohort had continuity of care, meaning over 50 percent of PCC visits were to the same PCC [27]. Coordination of care between an individual's oncology team and their PCC, as well as the coordination of care between the UCC clinic and PCCs, could be the focus in a future study. Additionally, individuals living with cancer may be seeing their PCC to provide updates on the status or progress of their

cancer care, suggesting there is actually more fragmented care. In recent years, PCCs are becoming more involved during treatment by helping to manage treatment side effects, co-existing chronic conditions, and psychological distress [37]. Future qualitative studies should examine the reason for PCC visits, as well as the quality of PCC visits during the treatment phase of care.

There was a 7% non-significant increase in hospitalizations after the opening of the UCC clinic. It is possible that individuals diagnosed with cancer who are experiencing severe symptoms are still hospitalized as a result of needing higher acuity care that is provided in a hospital setting. The slight increase may be explained by other things happening in the health care system during this time period. For example, 2014-15 experienced a particularly bad influenza season [38]. Individuals diagnosed with cancer are often immunocompromised and are susceptible to developing severe complications from infections such as influenza [39], which may explain the increase in hospitalizations during 2014-15. Furthermore, there has been an increase in the available treatments (i.e., Ibrutinib for chronic lymphocytic leukemia) offered in Manitoba, possibly exposing individuals to the potential for greater toxicity issues, resulting in increased hospitalizations [40].

After the implementation of the UCC clinic, there was a non-significant 4% decrease in ED visits overall and a non-significant 3% increase in ED visits during the UCC clinic hours of operation. The largest changes occurred at the two major EDs in Winnipeg. The HSC saw a 32% increase in the rate of ED visits, while the St. Boniface Hospital experienced a 31% decrease in the rate of ED visits during the UCC clinic hours of operation. There was minimal impact on other EDs and no significant trends by cancer type or CTAS score. Nearly 10% of the individuals with an ED visit at the HSC during the UCC clinic hours of operation also had a visit to the UCC clinic on the same day. However, this was not seen at the other EDs in Winnipeg. Therefore, the UCC clinic may act as a "screening" clinic for individuals who are sick enough that they should be seen by an ED but are first assessed at the UCC clinic. These individuals would be sent to the closest ED, which is at the HSC. This may partially explain the increase in the rate of ED visits seen at the HSC. The decrease in ED visits at St. Boniface Hospital may also contribute to the increase in ED visits at the HSC. Also, the number of individuals diagnosed with cancer in Manitoba is increasing [41]; hence, the population at risk of needing UCC clinic care is also likely increasing. However, the capacity of the UCC clinic has not increased over the years. This may result in individuals being sent to the ED when the UCC clinic capacity has been reached. In our previous study [27] examining predictors of ED and UCC clinic use, we found that ED use was highest immediately after diagnosis, whereas UCC clinic visits increased over the first four months after diagnosis. This highlighted the need to provide education on the UCC clinic to individuals at the time of diagnosis. Receiving chemotherapy was the strongest predictor for going to the UCC clinic; however, a recent history of using the ED was among the strongest predictors of going to the ED. It may be that some individuals go to the ED because that is what they are used to doing. Since the capacity of the UCC clinic has not changed since its inception, it is possible that the UCC clinic is not reaching enough people to see a substantial drop in other health care utilization. The results of our prior research, along with the current study, can be used to target individuals in the period immediately after diagnosis in order to increase UCC clinic visits in this time frame [27].

The current study examined the impact of the UCC clinic on ED visits, PCC visits, and hospitalizations. Based on the data available at the time, we were unable to look at the wait times for individuals who were sent to the ED from the UCC clinic. It was hypothesized that when individuals were sent to the ED from the UCC clinic, they received more rapid attention and stabilization at the UCC clinic first, resulting in lower wait times and a better patient experience in the ED. Unfortunately, we did not have the data to evaluate this hypothesis within the current study. This would be an interesting focal point for future research. Additionally, one of the current operational aims of the UCC clinic in its contemporary form is presumably reduced use of the individual's main clinic for urgent

issues in order to preserve space for stable planned outpatient and follow-up visits. A future study could evaluate the impact of the UCC clinic on the requests for urgent CCMB clinic visits. Individuals with urgent issues showing up in the main CCMB clinics can be sent to the UCC clinic to prevent delays in care for other individuals waiting in the main clinic. Therefore, the impact of the UCC clinic on the time spent waiting in the main CCMB clinic could be the focus of a future study. Furthermore, the cost-effectiveness of the UCC clinic could be evaluated in a future study.

Although prior research has emphasized the impact of urgent cancer care on the health care system, using outcomes that measure health care use may not adequately explain the benefits of an UCC clinic for individuals diagnosed with cancer. Outcomes such as patient quality of life and patient and health care provider experience may be more appropriate. Research in this area is limited, with prior studies in other jurisdictions stating that there is high patient and provider experience with UCC clinic models. Many individuals report preferring to attend a UCC clinic model of care rather than an ED [25,26]. Moreover, our prior research [27] found that the average wait time in the UCC clinic at CCMB was much shorter compared with an ED (2 h versus 9 h, respectively). The UCC clinic at CCMB may make the cancer journey less complicated and likely meets a role that, prior to its implementation, went unfulfilled in the health care system. This role could more accurately be described as a rapid or quick access clinic rather than an urgent care clinic. Additionally, this role might complement the care provided by the individual's oncologist, PCC, and traditional EDs, leading to more comprehensive patient-centered care, resulting in a better patient experience. Future qualitative research is needed to better understand the impact of the UCC clinic at CCMB on cancer care and individuals diagnosed with cancer.

### 4.2. Comparison with Previous Studies

Prior research is limited and has shown conflicting results regarding how the implementation of a UCC clinic impacts other health care utilization. The implementation of an ED designed specifically for individuals diagnosed with cancer in Seoul, Korea resulted in decreased admissions to inpatient units. The pre-post evaluation found that 42% of individuals were admitted to an inpatient unit after the specialized ED opened, a significant reduction from 85.5% admitted to inpatient units prior to the implementation [15]. However, the ITS evaluation of a UCC clinic established at a cancer center in Texas did not find any significant changes to hospitalizations after the UCC clinic opened. In this study, the authors concluded that higher acuity patients were still needing to be hospitalized [20], which is similar to our findings regarding hospitalizations in Manitoba. The same study evaluating a UCC clinic in Texas found a significant reduction of 15.3 fewer monthly weekday ED visits after the UCC clinic opened [20]. Similarly, the pre-post evaluation of an OECC in Connecticut found a non-significant decrease in hospitalizations and a significant decrease in ED visits of 4.6 per 100 patients after the opening of the OECC [22]. In Pennsylvania, a cancer symptom management clinic within a cancer center was implemented for individuals undergoing chemotherapy treatments. This study used a pre–post design and found that ED visits increased after the opening of this clinic from 34.5 to 38 median monthly visits. One of the conclusions in this study was the need to improve education on how to manage symptoms and coordination of supportive services [21]. Comparing the impact of UCC clinics across jurisdictions must be conducted with caution, as each UCC clinic is unique in the design of the clinic, the setting and location of the clinic, the individuals served by the clinic, and the study design and methods used to evaluate the clinic.

### 4.3. Strengths and Limitations

The current study's results add to the literature by rigorously evaluating the impact of the UCC clinic at CCMB on other health care utilization. Prior studies have examined the impact of a UCC clinic on ED visits and hospitalizations but have not looked at PCC visits. The current study addressed this gap in the literature. This study used data from population-based administrative health databases that have been previously validated

for accuracy [29–31]. As such, there were low numbers of missing data in our study. Additionally, we used an ITS study design [28] with a long baseline period, which allowed the trends of ED visits, PCC visits, and hospitalizations prior to the UCC clinic opening to be accounted for in the analysis. Pre–post designs, which are used by many of the previous studies evaluating UCC clinics, did not account for baseline trends. Not accounting for these historical trends can underestimate or overestimate changes in health care utilization after the implementation of the UCC clinic. Also, we included a seasonality term in order to account for seasonal fluctuations in health care use. One of the assumptions of using an ITS design is that there are no other changes occurring in the health care system. We cannot guarantee this was the case, so the results we saw may not be a direct reflection of the UCC clinic opening. We found statistically significant differences between individuals diagnosed before and after the opening of the UCC clinic in terms of cancer type and receiving systemic therapy or surgery within six months of a cancer diagnosis. There were 18,800 individuals in the cohort, which is a large sample size, so even small differences will be found to be statistically significant. For cancer type, the major differences were for breast and lung cancer. Before the opening of the UCC clinic, 18.5% and 12.3% of individuals were diagnosed with breast and lung cancer, respectively, whereas 16.9% and 13.7% were diagnosed with breast and lung cancer after the opening of the UCC clinic, respectively. Prior to the UCC clinic opening, 42.7% of the cohort received systemic therapy within six months of diagnosis and 46.3% of the cohort received systemic therapy after the UCC clinic opened. Similarly, 53.9% of individuals received surgery within six months of diagnosis before the UCC clinic and 47.9% of individuals received surgery after the UCC clinic was implemented. These differences are relatively small when working with a large sample size but are important to note as other possible explanations for the results found in this study.

In the final analysis, we included ED visits, only ED visits during UCC clinic hours of operation, PCC visits, and hospitalizations in the six months after an individual's cancer diagnosis. We also conducted a sensitivity analysis looking at the outcomes within one year after an individual's cancer diagnosis and did not find any significant changes to the results. In the stratified analyses by cancer type, ED location, and CTAS score, sample sizes may not have been large enough, resulting in a power issue and the inability to detect significant changes in health care utilization. To be consistent in the analysis, we limited the study to individuals who lived in the City of Winnipeg at the time of diagnosis; however, the UCC clinic does provide care to individuals living outside Winnipeg. Therefore, this study did not provide information for these individuals. On some occasions, the UCC clinic closed early due to capacity issues, so some individuals would not have been able to be seen at the UCC clinic and would have had to go to the ED instead. Similarly, if the duration of care required by an individual was estimated to be beyond the time when the UCC clinic closed for the day, arrangements for referral to an ED would be made. At the time of analysis, we had two years of UCC clinic data available. However, changes to the structure of the UCC clinic have since been made. Therefore, a future study should expand the years of the study to see whether there has been an impact on other health care utilization in the current state.

## 5. Conclusions

The UCC clinic at CCMB was implemented so that individuals diagnosed with cancer and serious blood disorders who were experiencing complications from the underlying disorder or its treatment could receive cancer-specific care from trained health care providers in a timely fashion. Even though there was minimal impact on ED visits, PCC visits, and hospitalizations in this study, it is believed that the UCC clinic may complement the care provided by the individual's oncologist, PCC, and traditional EDs, leading to more coordinated patient-centered care. However, this needs to be assessed in future studies. The benefit of the UCC clinic may be better seen by looking at the impact of the UCC clinic on other CCMB clinical operations (i.e., requests for urgent appointments within an individual's main clinic or the time spent waiting in the regular CCMB clinics), which should be the focus of a future study. It may be that health care utilization measures do not

capture the true impact of the UCC clinic, so future qualitative work should be completed to examine the impact of the UCC clinic on the quality of life of individuals diagnosed with cancer and patient and health care provider experience.

**Supplementary Materials:** The following supporting information can be downloaded at: https://www.mdpi.com/article/10.3390/curroncol30070496/s1, Figure S1. Rate (per person-days) of emergency department visits during the urgent cancer care clinic hours of operation for (a) CTAS scores 1 to 2 and (b) CTAS scores 3 to 5 by month, Winnipeg, Manitoba; Figure S2: Rate (per person-days) of emergency department visits during the urgent cancer care clinic hours of operation for (a) breast cancer, (b) digestive cancers, (c) lung cancer, and (d) genitourinary cancers by month, Winnipeg, Manitoba; Table S1: Cancer type categories; Table S2: Ratios and 95% confidence intervals between fitted and counterfactual values.

**Author Contributions:** Conceptualization E.J.B., P.C., T.F., B.G., M.K., H.S. and K.D.; methodology, K.G., P.L., O.B. and K.D.; formal analysis, K.G., P.L. and O.B.; data curation, K.G., P.L. and O.B.; writing—original draft preparation, K.G. and K.D.; writing—review and editing, K.G., P.L., E.J.B., P.C., T.F., B.G., M.K., H.S., O.B. and K.D.; visualization, K.G., P.L., O.B. and K.D.; supervision, K.D.; project administration, K.D.; funding acquisition, P.L., P.C., M.K., H.S. and K.D. All authors have read and agreed to the published version of the manuscript.

**Funding:** This research was funded by the Canadian Institutes of Health Research (A02-151563).

**Institutional Review Board Statement:** The study was conducted in accordance with the Declaration of Helsinki, and approved by the University of Manitoba's Health Research Ethics Board (project code: HS20816 (H2017:167), approval date: 24 May 2017), Manitoba Health's Health Information and Privacy Committee (project code: 2017/2018–08, approval date: 3 October 2017), CancerCare Manitoba's Research and Resource Impact Committee (project code: 2017-13, approval date: 15 June 2017), and Winnipeg Regional Health Authority's Research Access and Approval Committee (now known as Shared Health's Approval Committee for Privacy, Impact and Access in Research) (project code: 2017-044, approval date: 15 September 2017).

**Informed Consent Statement:** Patient consent was waived because data were de-identified.

**Data Availability Statement:** The data that support the findings of this study are not publicly available to ensure and maintain the privacy and confidentiality of individuals' health information. Requests for data may be made to the appropriate data stewards (Manitoba Health, Seniors and Active Living's Health Information Privacy Committee, CancerCare Manitoba's Research and Resource Impact Committee, and Shared Health's Approval Committee for Privacy, Impact and Access in Research).

**Acknowledgments:** We gratefully thank the Canadian Institutes of Health Research for their support as well as CancerCare Manitoba, Manitoba Health, Seniors and Active Living, and the Winnipeg Regional Health Authority for the provision of data.

**Conflicts of Interest:** The authors declare no conflict of interest. The funders had no role in the design of the study; in the collection, analyses, or interpretation of data; in the writing of the manuscript; or in the decision to publish the results.

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
