# Peer review of "Evaluation of the Impact of the Urgent Cancer Care Clinic on Emergency Department Visits, Primary Care Clinician Visits, and Hospitalizations in Winnipeg, Manitoba"

_curroncol, doi:10.3390/curroncol30070496_

Round 1
Reviewer 1 Report
Dear Authors,
The manuscript is overall well written. The aim of this work is to describe the results of the evaluation of the Impact of the Urgent Cancer Care Clinic (UCC) on Emergency Department (ED) Visits, Primary Care Clinician Visits (PCC), and Hospitalizations in Winnipeg.
The manuscript is very descriptive, the authors need to explain What is the new knowledge or value added of the study?
An interrupted time series study design was used to compare the rates of visits from January 1, 2010 to December 31, 2015. The center opened on November 4, 2013.
Are there more centers with these characteristics (UCC clinic)? for managing complications of the underlying diagnosis or its treatment in individuals diagnosed with cancer in a prompt manner. In Canada, EEUU or Europe? What methodology has been used to analyze its operation?
Line 64. Information about the impact of UCC clinics on other health care utilization is limited. Prior research has shown that emergency department (ED) visits and hospitalizations may decrease after the implementation of an UCC clinic, but the results are not consistent across jurisdictions [1,5–7].
OK, but what numbers come out. Nor is the impact significant in other locations? Authors write References in the literature, but numerical data and geographical locations are missing.
Line 69. The objective of this study was to evaluate the impact of the UCC clinic at CCMB on 69 ED visits, PCC visits, and hospitalizations in Winnipeg, Manitoba, Canada from November 2013 to December 2015.
What has happened from 2015 to 2019, is there no more data? Has a validation been done? A decade of study 2010-2020 would be better (although in 2020 we had COVID)
Future qualitative work should focus on the impact of the UCC clinic on patient quality of life and patient and health care provider satisfaction.
What bibliography is there about it, have other similar centers analyzed it?
Line 109. Manitoba Cancer Registry (MCR) was used to identify individuals diagnosed with invasive cancer (excluding non-melanoma skin cancer) between 2009 and 2015 who were living in Winnipeg, 18 years of age or older at the time of diagnosis, and were receiving treatment or follow-up care at CCMB.
Line 173. The results describe the number of PCC visits, ED visits, etc.,
How many patients are the authors referring? to know the number of patients, but there is no cohort of patients with cancer data, and clinical characteristics, sex differences,
Rates of ED visits were also stratified by cancer site.
Which method? How did the authors classify/represent cancer site? The results showed grouped cancers.
Line 135, What about other types of cancer? Why only breast, digestive, lung, genitourinary, and hematologic cancers. On what bibliography is it based to classify? The results could also be analyzed according to the severity of the urgency.
Line 227 The largest changes occurred at the two major EDs in Winnipeg. The HSC saw a 32% increase in the rate of ED visits, while the St. Boniface Hospital experienced a 31% decrease in the rate of ED visits during the UCC clinic hours of operation.
It seems that depending on the hospital location, the data obtained are contradictory
Line 230. There was minimal impact on other EDs and no significant trends by cancer site
Line 249. There was a 7% non-significant increase in hospitalizations after the opening of the UCC clinic.
The authors explain and interpret them, by complications from infections such as influenza, etc . However, this would further justify the fact of extending the study interval to years after 2015, or validating it in another range of years, to see if the behavior is maintained or varies.
The references are generally quite old. Only 4 of 19 references from the last 5 years (2018 onwards)
The raw data of the investigation have been deposited? I think this could be done even working with patient data anonymized?
I wish that these comments could help the authors to improve the paper.
Reviewer 2 Report
Thank you for asking me to review this article. It is a well-designed ITS study assessing the associated between UCC and ED visits, PCC and hospitalizations. The authors found no significant change in these outcomes after UCC implementation.
Abstract:
The mention of the word non-significant followed by the % change for the outcomes in the abstract is a bit misleading. I would include the 95% CI or just leave as non-significant without any % change. The non-significant word is not used with PCC. Is this significant or not? I am not sure I was able to find it in the body of the results either. Can the p-values be included for transparency? The omission of the word non-significant for the PCC outcome when the 95% CI is 1-1.13 seems to be a bit misleading as this is the only outcome without the specification of non-significant. What is the p-value. The conclusions do not fit with the data presented. There is no data to support the conclusion that the UCC complements the care provided by the individual’s oncologist, PCC and traditional EDs leading to more coordinated patient-centered care.
Introduction:
The first paragraph of the introduction represents methods more so than introduction. I would move it to the methods. I would also suggest the authors use a checklist to describe their UCC intervention. It doesn’t sound like they were directly involved in its original implementation. Still, it would be helpful if they could describe it in more detail following something like the TIDieR checklist (Hoffmann, T. C., Glasziou, P. P., Boutron, I., Milne, R., Perera, R., Moher, D., Altman, D. G., Barbour, V., Macdonald, H., Johnston, M., Lamb, S. E., Dixon-Woods, M., McCulloch, P., Wyatt, J. C., Chan, A.-W., & Michie, S. (2014). Better reporting of interventions: Template for intervention description and replication (TIDieR) checklist and guide. BMJ, 348(mar07 3), g1687–g1687. https://doi.org/10.1136/bmj.g1687).
In the introduction, it would be helpful for readers to understand the demand and needs that the UCC sought to fill with its implementation. This could include an overview of the rates of complications related to cancer and blood disorder treatments and the associated consequences of this (e.g., morbidity, mortality, healthcare utilization, costs).
Results:
The first sentence in the results should be moved to methods. Can a Table 1 be included with regards to the patient demographics who sought care at the UCC? Also, it would be helpful to see the number of patients being seen at UCC over the study period. Were there any changes in the patient population being seen at the UCC over the study period? As you mentioned in the discussion, new treatments are being used in cancer patients. Is it possible that a new treatment came out and that more patients with that specific cancer type receiving that treatment increased?
It might be helpful to place subheadings in the results so that we can follow each outcome (e.g., PCC, ED)
For the figures, is there a unit of measure for the rates on the y axis? They are hard to interpret at the moment partly for this reason.
Page 7 line 211-216 seems out of place in the results. If you include a demographic/UCC cancer visit table as the first section of the results to describe to readers the context before and after implementation, it may be possible to move these sentences to that location in the results.
Discussion:
Page 7 line 238 to 240 re: the number of cancer patients increasing – any Manitoba data to support this? Are these patients the same patients that would be accessing UCC? We know that there are more and more patients being diagnosed with early-stage cancers. Are these patients equally likely to use the UCC? I think if you are going to include this argument in the discussion than you need to discuss all facets of it and support it with data.
Page 7 line 242 you mentioned an increase in PCC visits and your main statement is that it is helping coordinate care. There are many different reasons why this may be associated with PCC increase use. This includes things like “doctor shopping” if patients are not satisfied with the care provided. Alternatively, in some situations, physicians meet with patients because they need to get updates about the any recent progress in their care. So in reality, it could also be a result of more fragmented care. I would try to ensure you are presenting all facets of the arguments you use to tell the story in the results.
It would be helpful to include alternatives to UCC that can be used to improve the quality of care related to cancer and its complications. This includes things like self-management of these issues with support over the phone if required. These alternatives that could be used in addition or replace UCC should be discussed to provide a balanced discussion.
There are a few key papers on emergency service utilization in cancer patients that could be referenced it not already referenced:
Lash, R. S., Bell, J. F., Reed, S. C., Poghosyan, H., Rodgers, J., Kim, K. K., Bold, R. J., & Joseph, J. G. (2017). A Systematic Review of Emergency Department Use Among Cancer Patients. Cancer Nursing, 40(2), 135–144. https://doi.org/10.1097/NCC.0000000000000360
Lash, R. S., Hong, A. S., Bell, J. F., Reed, S. C., & Pettit, N. (2022). Recognizing the emergency department’s role in oncologic care: A review of the literature on unplanned acute care. Emergency Cancer Care, 1(1), 6. https://doi.org/10.1186/s44201-022-00007-4
Round 2
Reviewer 1 Report
All comments have been addressed
Author Response
Thank you for taking the time to review this paper again.
Reviewer 2 Report
Thank you for your thoughtful comments and edits. The paper is significantly improved. Below I have included some additional comments for your consideration:
ABSTRACT
Line 26-28: The Urgent Cancer Care (UCC) clinic at CancerCare Manitoba opened in 2013 to provide 26 timely and contextual care to individuals diagnosed with cancer and serious blood disorders expe- 27 riencing complications from the underlying disorder or its treatment. This study examined the im- 28 pact of the UCC clinic on other health care utilization in Winnipeg, Manitoba, Canada. It is not clear why you have picked healthcare utilization as an outcome based on the fact that UCC were implemented to solve the problem of “timely and contextual care”. I would remove this phrase “timely and contextual” as it is extremely vague. I would directly link the problem that UCC clinics were implemented to solve with the outcomes that you sought to measure.
Abstract line 32-34: “We found a non-significant 4% decrease in the rate of ED visits, a 3% non-significant increase in the rate of ED visits during the UCC clinic hours, a significant 6% increase in PCC visits, and a non-significant 7% increase in hospitalizations after the UCC clinic opened.”I think the % changes need to be included as % change (95% CI, p-value) and not how it is currently written. For example, we found a 4% (95% CI, p-value) decrease in the rate of ED visits, a 3% (95% CI, p-value) increase in the rate of ED visits during the UCC clinic hours, a 6% (95% CI, p-value) increase in PCC visits, and a 7% (95% CI, p-value) increase in hospitalizations after the UCC clinic opened.
Abstract line 35-38: The author’s conclusion is still too strong for the data presented in the abstract (see Chiu, K., Grundy, Q., & Bero, L. (2017). ‘Spin’ in published biomedical literature: A methodological systematic review. PLOS Biology, 15(9), e2002173. https://doi.org/10.1371/journal.pbio.2002173). Suggest the following: “The implementation of UCC had minimal impact on health care utilization. Future work should aim to assess the impact of the UCC on other aspects of healthcare utilization (e.g., time spent in ED, number of tests ordered, cost effectiveness, etc.) and patient quality of life and patient and health care provider EXPERIENCE.” Note that I changed satisfaction to experience as this is a more comprehensive term (see Alessy, S. A., Alhajji, M., Rawlinson, J., Baker, M., & Davies, E. A. (2022). Factors influencing cancer patients’ experiences of care in the USA, United Kingdom, and Canada: A systematic review. EClinicalMedicine, 47, 101405. https://doi.org/10.1016/j.eclinm.2022.101405)
INTRODUCTION
Line 63-73: The introduction is inadequate for setting up your study. Overall, I think the introduction would be best structured as follows:
Paragraph 1: Experiences of people living with cancer and make it very clear to your reader what portion of the cancer journey you are describing. For example, it is likely that you will end up focusing on acute symptoms and complications of cancer and its treatment during the period where they are receiving treatment. You should finish here on making readers understand why a solution is needed to improve the acute care needs during the phase of cancer treatment (I assume there is evidence to support that this period in the cancer journey is associated with one of the higher rates of healthcare utilization but might be good to find something that states this)
Paragraph 2: Review of described solutions to improve the “value of healthcare” during the phase of active cancer treatment in the cancer journey - meaning improved results, quality and costs. Here you should set up the reader up for why UCC was the selected intervention in Winnipeg. Maybe it is the only described solution. Maybe it was the most realistic for the context of the healthcare system in Manitoba. But we need to know why UCC was the selected intervention for the identified problem. Line 83-89 sort of speaks to this but I think this needs to be stated more as a hypothesis… for example “A UCC clinic opened at Cancer- 83 Care Manitoba (CCMB) in November 2013. It was hypothesized that since the UCC clinic at CCMB is located 84 within CCMB’s MacCharles Unit, where its clinicians have access to the patient’s oncology 85 team and CCMB’s electronic medical record, and follow up care after an UCC clinic visit 86 may be provided by an individual’s primary care clinician (PCC), the UCC clinic offers 87 the unique opportunity to provide more coordinated care between an individual’s oncol- 88 ogy team and their PCC.” But again.. You are not really testing this hypothesis with your study.. Can you reframe with in a way that hypothesizes that UCC would improve the problem that you have set out to measure in more direct terms? I think we can probably extrapolate from what you are stating here that the outcome measures you have selected are appropriate but it feels like too big of a leap for the average reader with limited implementation knowledge.
Paragraph 3: Review of the gaps in what is known about the implementation of UCC. Here, you can use some of the text from line 95-102. Overall, I think you need to structure your assessment of the literature and evidence around the evaluation of the UCC using something like the REAIM framework for evaluating interventions (https://re-aim.org/learn/what-is-re-aim/). This can help you speak to the gaps in the literature and why you selected to study these outcomes.
Here is some feedback for the current introduction:
Your first paragraph discusses only ED visits. Suggest focuses on the overall patterns of healthcare utilization and the reasons for this rather than focuses on ED visits only. Also you are not specific enough with your references to the cancer journey as the reasons for healthcare utilization vary across the periods of diagnosis, treatment and survivorship for example. I recommend being very clear on which parts of this you are wanting to focus on for this paper and clearly defining that for your readers. For example, you state “Individuals who are diagnosed with cancer experience higher emergency depart- 63 ment (ED) use than individuals who do not have cancer.” … This makes it sound like you are referring to the period of time when they are initially presenting with symptoms and being diagnosed and starting treatment. But then you go on to say “This is a result of needing to seek 64 care for symptoms related to their cancer diagnosis and its treatment [1-3].” So its not concordant… and then you flip back to those being diagnosed… “The most common reasons for attending an ED among individ- 67 uals diagnosed with cancer are pain, fatigue, respiratory complaints, gastrointestinal com- 68 plaints, fever, and other infection-related issues [1,4,7-13].” Are you referring to symptoms leading to diagnosis?
Line 69-76: “It is estimated that over half of ED visits among individuals diagnosed with cancer are preventable with proper symptom management [5,9,14]. Furthermore, EDs often lack cancer-specific resources needed to support this complex population of individuals. Therefore, EDs may not always be the most appropriate setting to provide care for individuals diagnosed with cancer [5,13]. In recent years, urgent cancer care (UCC) clinics within cancer centres have become popular alternatives to traditional EDs for individuals diagnosed with cancer who are experiencing acute complications related to the underlying cancer diagnosis or its treatment [15-26].”
I don’t think this sets your study up well for the reader as it is suggesting that UCC should/could be used as an ALTERNATIVE to ED. However, in your study, it does not seem to be the mechanism of action as there was no change in ED visits. It appears that the two may be complementary? I am not sure.. But your results do not support that UCC are used purely as an alternative or substitute to ED. I think setting up the gap in the literature here would strengthen what your paper adds to the body of evidence. For example, “In recent years, urgent cancer care (UCC) clinics have been implemented in major cancer centers for individuals diagnosed with cancer who are experiencing acute complications related to the underlying cancer diagnosis or its treatment. However, the impact of UCC is not well understood.” I would recommend line 77-88 be deleted or replaced with actual references to either qualitative or quantitative studies supporting claims that “UCC clinics are able to provide comprehensive and contextual care to individuals 77 diagnosed with cancer since the clinics are usually staffed by health care providers with 78 specialized training in oncology.” and “An UCC clinic also provides in- 81 creased convenience and familiarity for individuals diagnosed with cancer and their fam- 82 ilies which may result in fewer delays in seeking care.” You have references here “UCC clinics have been shown to provide timely and safe 79 care and play an important part in managing acute conditions related to cancer-related 80 symptoms and treatment complications [15,16,19,27].” I would include the actual numbers from these studies that show support this very generalized claim.
Line 95-97: “The im- 95 plementation of an ED designed for individuals diagnosed with cancer in Seoul, Korea 96 resulted in a significant reduction in admissions to in-patient units” … Is an ED designed for individuals diagnosed with cancer the same thing as UCC?
Line 100: “Similarly, an Oncology Extended Care Clinic (OECC) in …” Same comment as above.. Is this the same thing as a UCC? What are the differences here if any?
Line 104-106: “Since 104 each UCC clinic is different in the way the clinic is designed, where the clinic is located 105 and, the individuals seen by the clinic, it is important to understand the impact of the UCC 106 clinic on health care utilization in each unique setting.” While this is true… it also lends itself to the question of why even publish this then if the results are only applicable to one unique setting… can you also add something that speaks to why this is also important for a broader audience… maybe it is similar healthcare settings can learn from it? If that is the case then I think you should provide more information on the unique aspects of Winnipeg and Manitoba so readers can understand how these results apply to them..specifically geography and aspects of the public healthcare system
METHODS
Line 136-137: …provide timely, comprehensive, coordinated and contextual care for 136 individuals diagnosed with cancer and serious blood disorders who are experiencing 137 complications of the underlying disorder or its treatment…. This feels to vague. Can you provide in more concrete terms what the solution they were trying to solve was and how this is linked to the outcome you are trying to evaluate?
Line 213: cancer sites is an odd terminology as it can be mistaken for a physical location.. I would refer to it as cancer type or cancer diagnosis .. please change throughout entire manuscript
Line 148, line 183-185 & line 203-204: Patients must be receiving treatment or follow up care to have access to UCC but you included patients diagnosed with cancer in the last 6 months. What defines “diagnosis”? Is this date of pathology or date of consultation in the cancer center? Why was this time period selected? Not all patients diagnosed with cancer are on treatment within 6 months? Even less are on follow up.. So how many actually had access to UCC here?
Line 215-217: “A sensitivity analysis was conducted that included all ED visits, only ED visits 215 during the UCC clinic hours of operation, PCC visits, and hospitalizations during the first 216 year after an individual’s cancer diagnosis.” I do not see these sensitivity analyses reported in the results? Maybe I missed it..
Line 209-214: “Rates of ED visits during the 209 UCC clinic hours of operation were also stratified by ED location (HSC, St. Boniface Hos- 210 pital, Seven Oaks General Hospital, Grace Hospital, Concordia Hospital, and Victoria 211 General Hospital), CTAS score (1 to 2 and 3 to 5), and cancer site (breast, digestive, lung, 212 genitourinary, and hematologic cancers). These cancer sites were grouped to ensure large 213 enough sample sizes for the analysis and were chosen based on discussions with clinicians 214 (Table S1).” These are subgroup analyses? I would make sure all the sensitivity and subgroup analyses are discussed as such and the rationale for these analyses are clearly explained to the reader to avoid it looking like there was “fishing”.
RESULTS
Line 252-254: “The analysis of health care utilization was restricted to health care visits that occurred 252 within six months of diagnosis and only visits that occurred between 2010 to 2015 were included. Therefore, only individuals diagnosed between July 1, 2009 and December 31, 254 2015 contributed person-days to the analysis.” this should be in methods
Table 1: please include the proportion that were on treatment and what type of treatment they were receiving also include which ones were on follow up given that these are the two criteria for access to UCC . Please include p values here to compare the differences? Why not include p values?
Line 309-310: Are these the sensitivity analyses that you referred to in your methods? If so, why are they all included in the ED section? Suggest a new section for sensitivity and subgroup analyses as those are also included in the paragraphs that follow but do not have a heading so it looks like they are included in the ED section almost..
Line 343-346: There 344 was a non-significant decrease in ED visits during the UCC hours of operations for CTAS 345 scores 1 to 2 and a non-significant increase for CTAS scores 3 to 5…. I would stay away from this language of non significant vs. significant .. I would present the % change, 95% CI and p-values and let the reader conclude if this is meaningful (i.e., clinically and statistically significant). The focus on describing things as significant vs. non significant throughout the entire paper really negates the reader from making more nuanced conclusions for themselves. Please remove this type of language (i.e., “non significant” trend) from the entire paper and replace with the % change, 95% ci and p value.
DISCUSSION:
4.1 Main findings… The discussion needs to start with a general overview of the main results and what they add to the existing body of evidence on this topic before going into the subgroup analyses (i.e, UCC clinic hours of operation). For example, the fact that there was no meaningful change in healthcare utilization after implementation of UCC is a very important finding as this UCC is a huge effort for the healthcare system and the benefits of it need to be understood in order to continue to support this type of initiative! I think this is the crux of this paper. This initiative is a huge demand on resources without any proven benefit at this point….
Then I would make headings for each of the main outcomes and the subgroup/sensitivity analyses that were done e.g., starting with ED visits.. But these need to be in the same order as they are presented in the results and methods section.. The ed visits are presented last in the results section but first in the discussion? Can you mirror these please?
Line 356-358: “The 356 HSC saw a 32% increase in the rate of ED visits, while the St. Boniface Hospital experi- 357 enced a 31% decrease in the rate of ED visits during the UCC clinic hours of operation. 358” This is very important.. Were the EDs given commensurate additional resources to handle this increase as a result of UCC? People often work in silos in these situations but clearly this would be important to work with ED when implementing a UCC? Was this done?
Line 362-363: “Therefore, the UCC clinic may act as a “screening” 362 clinic for individuals who are sick enough that they should be seen by an ED but are first 363 assessed at the UCC clinic.” These makes UCC sound redundant and potentially unnecessary if there was otherwise no change in healthcare utilization… maybe its because there needs to be some changes to the way the UCC is staffed so they can actually offload the ED? Not sure.. But right now it sounds duplicative without any benefit..
Line 360: “Nearly 10% of the individuals with an ED visit at the HSC during the UCC clinic 360 hours of operation also had a visit to the UCC clinic on the same day.” Is this data presented in the results? I don’t see it? Maybe its in the supplementary somewhere… I would make sure to include all the data that you are highlighting in the discussion in the main results section and not in the supplementary.. Sorry if I missed it… Also, are you able to see how this 10% changed pre vs. post ucc implementation?
Line 367-371: “Also, the number of individuals diagnosed with cancer in Man- 367 itoba is increasing [1636]. Hence, so the population at risk of needing UCC clinic care is 368 also likely increasing., howeverHowever, the capacity of the UCC clinic has not increased 369 over the years. This, resulting may result in individuals being sent to the ED when the 370 UCC clinic capacity has been reached. 371” Can you provide the % increase during this time? Is it reflective of the % increase in ED? Maybe this is more of a limitation? Shouldn’t this be accounted for in the ITS analyses? Is there a way to account for this?
Line 374: “…with follow-up care provided by their PCC” How do you know this? Sometimes patients go see their PCC because they were not satisfied with the UCC?
Line 378-382: “A phenomenon 378 known as doctor shopping may be responsible for the increase in PCC visits. This occurs 379 when an individual sees multiple clinicians for the same illness [37]. However, our previ- 380 ous research showed that over 70 percent of individuals in the cohort had continuity of 381 care, meaning over 50 percent of PCC visits were to the same PCC [27] so it is unlikely 382 that doctor shopping is the reason for the increase in PCC visits.” I would argue that if someone is going to see another doctor for the same illness then that would also count as doctor shopping unless there is some evidence that the PCC was given instructions to manage the care moving forward… otherwise it could be that the patient was not satisfied with the UCC and then went to get an opinion from their PCC? If the UCC handled their concerns then there would potentially be no need to see the PCC if all the notes are sent to their PCC etc otherwise? Why are they visiting their PCC if they go to the UCC? This is probably an important area of further research as there is ample evidence to suggest that PCC do not feel like their is good care coordination with cancer centers in general…
Line 387-390: “In recent years, PCCs are becoming more involved during treatment by helping to 387 manage treatment side effects, co-existing chronic conditions, and psychological distress 388 [3817], resulting in more coordinated care between an individual’s oncology team and 389 their PCC.” The study you cite is assessing PCP visits 1-3 years after diagnosis which is a completely different period within the cancer journey compared to what you are studying. Furthermore it is quantitative data so it does not reflect anything about the purpose or quality of those visits. I think the question of reasons for the PCC visits needs to be discussed with much more “exploratory” language without any definitive conclusions one way or another but more likely an area for future study.
Line 394-396: Re: influenza .. you stated in your methods “Non-cancer related ED visits were excluded from 208 the analyses and are described in a previous paper..” Wouldn’t these have been excluded in this case?
Line 399: Re: treatments - any way to verify if there were changes in treatment during this period? This would be very helpful…
Line 428 & 430: Change satisfaction to experience
Line 505-513: “Even though, there was minimal impact on ED visits, PCC visits, and hos- 505 pitalizations in this study, it is believed that the UCC clinic plays an important role in an 506 individual’s cancer care and may act more as a rapid access clinic rather than an urgent 507 care clinic. The UCC clinic may complements the care provided by the individual’s oncol- 508 ogist, PCC, and traditional EDs leading to more coordinated patient-centered care. It may 509 be that health care utilization measures do not capture the true impact of the UCC clinic 510 so future qualitative work should be completed to examine the impact of the UCC clinic 511 on the quality of life of individuals diagnosed with cancer and patient and health care 512 provider satisfaction.” Again here there is too much Spin. Based on this study alone, we can see that the UCC is a huge use of resources with potentially limited resource savings in other areas. There may be more efficient ways to achieve the same outcomes… We have no evidence that it compliments care and may very well be duplicative! I think this is critically important to assess before further resources are put into sustaining something that may very well have no benefit. This view must also be presented here.
Round 3
Reviewer 2 Report
The authors have addressed all my comments very thoroughly. Thank you!